# Biochemical and Genomic Characterization of the Cypermethrin-Degrading and Biosurfactant-Producing Bacterial Strains Isolated from Marine Sediments of the Chilean Northern Patagonia

**DOI:** 10.3390/md18050252

**Published:** 2020-05-13

**Authors:** Patricia Aguila-Torres, Jonathan Maldonado, Alexis Gaete, Jaime Figueroa, Alex González, Richard Miranda, Roxana González-Stegmaier, Carolina Martin, Mauricio González

**Affiliations:** 1Laboratorio de Microbiología Molecular, Escuela de Tecnología Médica, Universidad Austral de Chile, Puerto Montt 5504335, Chile; carolina.martin@uach.cl; 2Laboratorio de Bioinformática y Expresión Génica, Instituto de Nutrición y Tecnología de los Alimentos, Universidad de Chile, Santiago 7810000, Chile; jomaldon@gmail.com (J.M.); alex.ignacio@live.com (A.G.); 3Center for Genome Regulation, Santiago 7810000, Chile; 4Laboratorio de Biología de Sistemas de Plantas, Departamento Genética Molecular y Microbiología, Facultad de Ciencias Biológicas, Pontificia Universidad Católica de Chile, Santiago 8331150, Chile; 5Instituto de Bioquímica y Microbiología, Facultad de Ciencias, Universidad Austral de Chile, Valdivia 5090000, Chile; jefigueroa@uach.cl (J.F.); rgstegmaier@gmail.com (R.G.-S.); 6Laboratorio de Microbiología Ambiental y extremófilos, Departamento de Ciencias Biológicas y Biodiversidad, Universidad de los Lagos, Osorno 5290000, Chile; alex.gonzalez@ulagos.cl; 7Escuela de Ingeniería Civil Industrial, Universidad Austral de Chile, Puerto Montt 5500000, Chile; rmiranda@spm.uach.cl; 8Laboratorio Medicina Traslacional, Instituto Clínico Oncológico, Fundación Arturo López Pérez, Santiago 8320000, Chile

**Keywords:** cypermethrin, biosurfactants, biodegradation capacities, marine sediments

## Abstract

Pesticides cause severe environmental damage to marine ecosystems. In the last ten years, cypermethrin has been extensively used as an antiparasitic pesticide in the salmon farming industry located in Northern Patagonia. The objective of this study was the biochemical and genomic characterization of cypermethrin-degrading and biosurfactant-producing bacterial strains isolated from cypermethrin-contaminated marine sediment samples collected in southern Chile (MS). Eleven strains were isolated by cypermethrin enrichment culture techniques and were identified by 16S rDNA gene sequencing analyses. The highest growth rate on cypermethrin was observed in four isolates (MS13, MS15a, MS16, and MS19) that also exhibited high levels of biosurfactant production. Genome sequence analyses of these isolates revealed the presence of genes encoding components of bacterial secondary metabolism, and the enzymes esterase, pyrethroid hydrolase, and laccase, which have been associated with different biodegradation pathways of cypermethrin. These novel cypermethrin-degrading and biosurfactant-producing bacterial isolates have a biotechnological potential for biodegradation of cypermethrin-contaminated marine sediments, and their genomes contribute to the understanding of microbial lifestyles in these extreme environments.

## 1. Introduction

Los Lagos region (S41°85′39″ W73°48′32″), located at the Chilean Northern Patagonia, has a high-density salmon farming industry with an extensive history of cypermethrin usage [1,2]. Cypermethrin [cyano-(3-phenoxyphenyl)methyl]-3-(2,2-dichloroethenyl)-2,2-dimethylcyclopro-pane-1-carboxylate is a synthetic pyrethroid pesticide used in agriculture and aquaculture [3], classified as a possible human carcinogen by the Environmental Protection Agency with moderate–acute toxicity according to World Health Organization [4,5]. This compound has an impact on marine ecosystems, affecting the biodiversity of fish and aquatic invertebrates [2,6]. Given its bioaccumulative effect and its further biomagnification in the food chain, the presence of this compound in sediment and soil water is of great concern [7]. As shown in a previous report [2], a high concentration of the pyrethroid cypermethrin, with values ranging from 18.0 to 1323.7 ng g^−1^, was observed in marine sediments in the Northern Patagonia, the same region from which our bacterial isolates were obtained. These habitats are extreme environments with a wide range of temperatures, ranging from 4 to 20 °C, and salinity (range 32%–33%), and also a low nutrient availability, as is the case for nitrate (NO_3_) and phosphate (PO_4_). 

Bioremediation has become an essential tool for removing these pollutants through biological methods, taking advantage of the degradative capabilities of microorganisms for cleaning contaminated environments [8,9]. The most recommended procedure is to use the local genetic resources, whose activity can be modified by modifying nutrients, water, air, and biosurfactants [9,10]. Natural degradation of cypermethrin is carried out by microorganisms that could produce the hydrolysis of the ester linkage, resulting in 3-phenoxybenzoic acid [11]. Diverse microorganisms isolated from soil have been reported to degrade cypermethrin, such as *Pseudomonas*, *Serratia*, *Streptomyces*, *Rhodobacter*, *Stenotrophomonas*, *Sphingomonas*, and *Bacillus* [12,13,14,15,16,17]. However, studies reporting cypermethrin-degrading bacterial strains obtained from marine environments are less frequent (i.e., *Cellulophaga lytica* DAU203) [18], especially from marine sediments.

Biosurfactant production increases the bioavailability of pollutants for microorganisms by increasing cell surface hydrophobicity [19]. This, in turn, improves the binding of organic compounds to the cell membrane, their adsorption, and ultimately, their biosorption [19,20]. Different microorganisms produce a diversity of biosurfactants [21]. However, there are only a few reported biosurfactant-producing microorganisms that have proven useful in cypermethrin biodegradation processes [4,22]. Therefore, the aims of this study were the identification and characterization of cypermethrin-degrading and biosurfactant-producing bacterial strains isolated from marine sediments in extreme environments, as is Chilean Northern Patagonia, and determining the presence of cypermethrin biodegradation pathway genes, integrating biochemical and genomics approaches. 

## 2. Results

### 2.1. Isolation and Identification of Marine Bacterial Strains

Eleven cypermethrin-degrading and biosurfactant-producing bacteria were isolated from cypermethrin-polluted sediment sampled at Manao Bay, Ancud, Chiloé (Figure 1a). Strains were isolated after three subcultures by enrichment with cypermethrin, with concentrations ranging from 50 to 200 mg L^−1^ (Figure 1b). According to Gram-staining analyses, five isolates were Gram-negative and six were Gram-positive. Strains were cocci, bacilli, and rod-shaped (Appendix A). All strains were catalase-positive, and five of them were motile. Four strains were capable of growth on 50 mg L^−1^ of cypermethrin as the sole carbon and energy source, while reaching the highest cell concentration measured by OD_600 nm_. These strains were selected for further analysis. The eleven marine bacterial strains (MS) obtained were compared using 16S rRNA gene identity; the results are summarized in Figure 1c. The phylogenic analysis showed that strains MS8, MS10, MS15a, and MS19 are closely related to *Pseudomonas*. MS1, MS11, MS12, MS13, MS14, and MS16 strains are closely related to *Rhodococcus*, and MS4 belongs to *Serratia* (Appendix A, Figure 1c). 

### 2.2. Surfactant-Producing Bacteria

The emulsion index (*E*_24_) produced by each of the eleven bacterial strains was quantified in order to identify the bacterial strain with the highest capacity of biosurfactant production. All the strains grown in the ETMS medium were capable of emulsion production, which was observed after the addition of diesel petroleum. Strains MS13, MS15a, MS16, and MS19 showed the highest emulsion indexes (>60%) among all the bacterial strains in these conditions (Appendix A). 

### 2.3. SEM Studies

The morphologies of selected marine bacteria grown until the mid-exponential phase were studied by scanning electron microscopy (Figure 2). MS13, MS15a, MS16, and MS19 strains showed cell pleomorphism with a rod, bacilli, and coccoid shapes. Cells usually occurred in pairs. In strain MS16, anchor-like appendages forming bridges from one cell to the next were observed (Figure 2). 

### 2.4. Genome Characterization of Selected Strains

To further investigate the cypermethrin catabolic pathway in selected marine bacterial strains, the whole genome was sequenced. The assembled genomes of *Rhodococcus* sp. MS13 and *Rhodococcus* sp. MS16 consisted of 6,460,280 bp and 6,970,856 bp, and 23 and 37 scaffolds, respectively; the average G + C content was 62% for both strains (Table 1). The assembled genomes of *Pseudomonas* sp. MS15a and *Pseudomonas* sp. MS19 consisted of 5,249,999 bp and 4,496,051 bp, and 27 and 30 scaffolds, respectively; the average G + C content was 65% and 57%, respectively (Table 1). The four sequenced genomes represent high-quality assemblies with N50 indexes over 290Kb, BUSCO completeness indexes over 96%, single copy proportions of BUSCO markers over 94%, and near 0% fragmented markers (Table 1).

### 2.5. Identification of Secondary Metabolites and Detection of Genes Involved in Cypermethrin Catabolism in MS13, MS15, MS16, and MS19 Marine Bacterial Strains, and Comparison with Reference Strains

In order to establish catabolic and anabolic capabilities of the four sequenced strains, an analysis of the biosynthetic coding capacity (BGC) using antiSMASH was performed, and the *estA*, *pytH*, and *laccase* genes encoding the cypermethrin catabolic pathway were identified. BGC’s analysis showed a differential pattern of bioactive compounds in all sequenced strains. Figure 3 describes the number of BGC detected in all studied strains compared with their respective reference strains using antiSMASH version 5.0. Of the four studied strains, *Rhodococcus* sp. MS13 had the highest number of gene clusters, with 17 BGC, followed by *Rhodococcus* sp. MS16, with 15 BGC. *Pseudomonas* sp. MS15a and MS19 had seven and five BGC, respectively. Compared with strains of the genus *Pseudomonas, Rhodococcus* showed a greater number of non-ribosomal peptide synthetases (NRPS). Regarding specific NRPS compounds, *Rhodococcus* sp. MS13 and *Rhodococcus* sp. MS16 presented siderophores, antifungals, and antibiotics, whereas *Pseudomonas* only had the first two. Figure 3 also describes the presence of genes involved in the cypermethrin degradation pathway and the biosurfactant compounds that MS13, MS15, MS16, and MS19 produce. Overall, laccases and carboxylesterases were detected in all analyzed genomes, but genes encoding esterases were identified only in *Pseudomonas* sp. strain (MS15a and MS19) genomes. 

The proposed cypermethrin degradation pathway for MS13, MS16, MS15a, and MS19 isolates is described in Figure 4. Genomic analysis suggests that the cypermethrin degradation pathway could be carried out by esterase, carboxylesterase, and laccase. 

## 3. Discussion

Since 1997, *Caligus rogercresseyi* has been recorded as the most serious parasite in the salmon industry in Northern Patagonia, Chile [1]. Several products have been used to keep sea lice under control in Chile. Cypermethrin [2] is one of them. As a consequence of the extended exposure time to the drug, Northern Patagonia presents high concentrations of cypermethrin in marine sediments [2]. In this scenario, some microorganisms may have acquired the cypermethrin catabolic pathway as a mechanism to use this organic carbon source in an environment with low availability of carbon and hard environmental conditions. 

In this study, cypermethrin-degrading and biosurfactant-producing bacterial strains isolated from marine sediment in Northern Patagonia were studied, along with their phenotypic, biochemical, and genomic properties, and their potential to bioremediate cypermethrin-polluted sediments. Based on 16S rRNA gene sequences analyses, bacterial isolates were related to the genera *Pseudomonas*, *Rhodococcus,* and *Serratia,* suggesting a lower abundance of bacterial genera associated with marine sediments in our sampling sites. Phylogenetic analyses indicate that these isolates are highly related to the reference strains *Rhodococcus erythropolis*, *Rhodococcus* sp., *Rhodococcus globerulus, Serratia odorifera*, *Pseudomonas oryzihabitans*, *Pseudomonas* sp., and *Pseudomonas marincola*. Isolates from the genus *Rhodococcus* were the most abundant. These results are consistent with those reported by Choi et al. [25], which indicated that *Actinobacteria* and *Gammaproteobacteria* were very prominent families in bacterial communities inhabiting the surfaces of such marine sediments [25]. 

Overall, clear morphological differences were observed between the eleven isolated strains; and all isolates were able to grow on cypermethrin as the sole carbon source and showed emulsion activity (biosurfactants). However, four of them (MS13, MS15a, MS16, and MS19) reached the maximum measurable growth on cypermethrin (measured by turbidity—optical density: OD_600_). Emulsions were also achieved in the cell-free extracts of these strains, indicating that the emulsifying activities were extracellular. Therefore, strains MS13, MS15a, MS16, and MS19 were selected for further characterization towards their application for cypermethrin bioremediation. 

Biosurfactants are amphipathic molecules with emulsifying and high surface activity; this turns them into attractive agents for bioremediation [26]. Strains MS13, MS15a, MS16, and MS19 were isolated from cypermethrin-contaminated sediment and showed high potential as biosurfactant-producing bacteria, reaching emulsion indexes of 77%, 60%, 79%, and 71% respectively. Techaoei et al. [27] reported biosurfactant-producing bacteria from soil samples that showed emulsifying activities ranging from 8% to 63% emulsification [27]. Borah and Yadav [28] reported that E24 tested with kerosene, crude oil, and engine oil reached lower values than the ones reported here; namely, 55%, 29%, and 20% emulsion indices, respectively [28]. Previously, it has been described that members of the *Rhodococcus* and *Pseudomonas* genera showed degrading activity of cypermethrin in polluted soils [10], along with *Serratia*, in in vitro assays [4], but to our knowledge, there are no studies on cypermethrin degradation in marine sediments. In this study we determined that, of all obtained isolates, four strains grew successfully in high cypermethrin concentrations, two *Rhodococcus* and two *Pseudomonas*. 

Therefore, these bacteria are good candidates for biodegradation of cypermethrin-contaminated marine sediments. Electron microscopy showed that marine strains usually occur in pairs and form bridges via a net-like structure. It has been reported that genera *Rhodococcus* and *Pseudomonas* have nocardioform, that is, rod-shaped or coccoid elements [29,30]. 

The results of the assembled genome of *Rhodococcus* sp. MS13 and *Rhodococcus* sp. MS16 are comparable with a bacterial strain isolated from marine sediment in Comau fjord, North Patagonia, by Undabarrena [31]. The isolated marine bacterium was *Rhodococcus* sp. H-CA and its complete chromosome has 6.19 Mbp with a content of 62.45% G + C. However, the presence of a *Rhodococcus* sp. RHA1 (RHA1) strain has also been reported, which has the largest bacterial genome sequenced to date, comprising 9,702,737 bp (67% G + C) [29]. Regarding the assembled genomes of *Pseudomonas* sp. MS15a and *Pseudomonas* sp. MS19, our genome data was comparable with strain *Pseudomonas* sp. S.C.T. isolated from marine sediment, whose genome was sequenced and assembled with 4.79 Mbp and 62.5% G + C content [32].

*Rhodococcus* sp. MS13 and *Rhodococcus* sp. MS16 have great numbers of biosynthetic clusters compared to the other two isolated bacteria; they also have the largest genomes—6.5 and 6.9 Mb. Both strains have great numbers of biosynthetic gene clusters in their respective genomes. These data are similar to those obtained by Ceniceros et al. [33], and Undabarrena et al. [31], which explored the metabolic capacities of the genus *Rhodococcus* and sequence the genome of *Rhodococcus* sp. H-CA8f [31,33]. The strains with the fewest gene clusters are *Pseudomonas* sp. MS19, which has the smallest genome size, 4.5 Mb, together with *Pseudomonas* sp. MS15, with seven gene clusters. The most represented BGCs correspond to non-ribosomal peptide synthetases (NRPS). These results show the biosynthetic potentials of the studied strains. 

In relation to NRPS, the strains studied presented siderophore, antifungal, and antibiotic bioactive compounds. Siderophores play an essential role in bacterial metabolism regarding iron uptake, since it has been described that there is a low concentration of iron in seawater (0.01–2 nM) [34,35]. It is interesting to note that *Rhodococcus* sp. MS13 contained more NRPS than *Rhodococcus erythropolis* PR4. One of them, glycinocin A, from *Rhodococcus* sp. MS16, generates lipopeptides, which can act like biosurfactants or antibiotics [34]; this strain showed the highest E24 index. 

The cypermethrin degradation pathway depends on three enzymes: esterase, carboxylesterase, and laccase [36,37,38]. Laccase is an enzyme involved in the oxidation of aromatic compounds, which should play a role in cypermethrin biodegradation. Gangola et al. [17] described a novel cypermethrin degradation pathway in *B. subtilis* [17]: *pytH* gene encodes a pyrethroid-hydrolyzing carboxylesterase capable of degrading a variety of pyrethroids, including cypermethrin, suggesting that these strains could be good cypermethrin-degrading candidates [36], although chemical analyses are yet to be done; cypermethrin and its intermediate metabolites should be analyzed because sometimes toxic intermediates metabolites such as 3-phenoxybenzoic acid exist. This one metabolite is a toxic intermediate of cypermethrin degradation, a recalcitrant chemical [13]. 

Cypermethrin is first degraded by esterases into 3-(2,2-dichloroethenyl)-2,2-dimethylciclopropane carboxylic acid and α-cyanogroup-3-phenoxybenzyl alcohol; the latter is transformed into 3-PBA [23,36]; cypermethrin can also to be degraded by carboxylesterases through hydrolysis of the carboxyl ester bond, resulting in 3-(2,2-dichloroethenyl)-2,2-dimethylciclopropane carboxylic acid and 2-hydroxy-2-(3-phenoxyphenyl) acetonitrile [24,36]. The former is converted into CO_2_ [36]. To select microorganisms good for bioremediation is necessary so that the cypermethrin-degrading strain can transform pyrethroid into non-toxic intermediate metabolites. These bacteria first could be used for in situ treatment of bioaugmentation in an environmentally-sustainable strategy to reduce cypermethrin levels in cypermethrin-contaminated sediment. According to Chun et al. [37], the stimulation of PCB-dechlorinating and degrading microorganisms with electron-donors/-acceptors addition contributed to the degradation of PCBs in sediment.

There is an approach that made the biodegradation of β-cypermethrin and 3-PBA more efficient—using a coculture of *Bacillus licheniformis* B-1 and *Aspergillus oryzae* M-4 [38].

The enzyme laccase uses molecular oxygen and a phenolic substrate for degrading cypermethrin [17,24,39]. *Laccase* and *pytH* genes were present in all strains, but the *estA* gene was present only in *Pseudomonas* sp., MS15a and *Pseudomonas* sp. MS19 (*in silico*). Amplification of cypermethrin-degrading genes in the selected bacterial strains revealed that the *pytH* and *laccase* genes were present in all studied strains (data not shown). Kubicki et al. [40], have described some biosurfactants produced by marine microorganisms [40]. Our results suggest that both *Rhodococcus* sp. synthesize viscosin, arthrofactin, and putisolvin; *Pseudomonas* sp. M15a synthetizes viscosin, amphisin, and putisolvin; and *Pseudomonas* sp. MS19 synthesizes rhamnolipid, amphisin and arthrofactin; this indicates that our studied strains have a variety of surface-active metabolites.

In summary, these four cypermethrin-degrading and biosurfactant-producing bacterial strains isolated from marine sediments have great biotechnological potential for biodegradation of cypermethrin-contaminated marine sediments.

## 4. Materials and Methods

### 4.1. Sampling Site

Sediment samples were collected in April 2018 from cypermethrin-contaminated sediments (*n* = 9). These samples were obtained in triplicate from three sampling sites, near a salmon farm in the Manao Bay, Ancud, located in the district of Chiloé (S41°51′16.6″ W73°29′05.6″), at 25 meter depth (Figure 1a). Samples were transported and stored at 4 °C until analyses. 

### 4.2. Culture Conditions

Bacteria were cultivated in Bushnell–Haas (BH) broth medium (composition per L: 1 g KH_2_PO_4_; 1 g K_2_HPO_4_; 1 g NH_4_NO_3_; 0.2 g MgSO_4_; 0.02 g CaCl_2_; and 0.05 g FeCl_3_ at pH 7.0), containing 50 mg L^−1^cypermethrin (Merck^®^) as the sole carbon source. Cypermethrin-degrading and biosurfactant-producing bacterial strains were grown in minimal ethanol salts medium (ETMS) (composition per L: 22.2 g K_2_HPO_4_ × 3H_2_O; 7.26 g KH_2_PO_4_; 0.2 g MgSO_4_ × 7H_2_O; 0.4 g (NH_4_)_2_SO_4_) and 25 mL of absolute ethanol (Merck^®^) [41]. Strains were also cultivated in Trypticase Soy Agar (TSA) (Difco^®^) and Trypticase Soy Broth (TSB) (Difco^®^). Cultures were grown at 28 °C in a gyratory shaker (model 3016A, Labtech) at 140 rpm. Cell growth was determined by optical density (O.D.) at 600 nm. 

### 4.3. Isolation of Bacterial Strains from Cypermethrin-Contaminated Marine Sediments

Using enrichment culture techniques, several bacterial species were isolated (Figure 1b). To do this, a 5 g sample of marine sediment was added to a 250 mL Erlenmeyer flask containing 50 mL of sterilized BH enrichment medium with an initial concentration of 50 mg L^−1^ of cypermethrin and incubated at 28 °C in a rotary shaker at 140 rpm for five days. After that, 5 mL of enrichment culture was inoculated into 50 mL of fresh BH medium containing 100 mg L^−1^of cypermethrin. Later, the enrichment culture was incubated five days in fresh enrichment medium containing 200 mg L^−1^ of cypermethrin. 

The final culture was serially diluted and spread on B.H. agar plates containing 50 mg L^−1^ cypermethrin for incubation at 28 °C for five days. To inhibit eukaryotic cell growth, 100 and 200 mg L^−1^ of cycloheximide (Sigma, St. Louis, MO, USA) were added to the first and second enrichments, respectively [11]. Plating was repeated until obtaining pure cultures and these were checked through Gram-staining using an Eclipse Ni-U (Nikon, Tokyo, Japan) optical microscope with 1000× magnification (Appendix A). Isolates were stored in glycerol (15%) and kept at −80 °C until use. Strains (Appendix A) were deposited in GenBank (MK271075, MK271077, MK271079, MK271080, MK271081, MK271082, MK271083, MK271084, MK271085, MK271086, and MK271087).

### 4.4. 16S rDNA Gene Sequencing

A colony of each isolated strain was obtained from the TSA medium, transferred to the TSB medium, and grown to OD_600nm_ = 0.6. Cells were harvested by centrifugation, and genomic DNA was extracted using the Wizard Genomic DNA Purification Kit (Promega, Madison, WI, USA), according to manufacturer’s recommendations. Universal primers 27F (5′-AGA GTT TGA TCA TGG CTC AG-3′) and 1492R (5′-CGG TTA CCT TGT TAC GAC TT-3′) were used to amplify 16S rDNA [42]. Master Mix containing *Taq* DNA polymerase (Promega, Madison, WI, USA) was used for PCR-amplification. PCR thermal cycling conditions for the amplification of 16S rDNA genes were: initial denaturation for 4 min at 94 °C, followed by 30 cycles of 30 s at 94 °C denaturation temperature; 1 min at 58 °C as annealing temperature; 1 min at 72 °C as extension temperature; and a final extension of 7 min at 72 °C. PCR products were visualized through agarose gel electrophoresis. PCR products from 16S rRNA amplification were sequenced by Macrogen (Seoul, Korea), using the primer 800R (5′-TAC CAG GGT ATC TAA TCC-3′).

### 4.5. Determination of Emulsion Index (E_24_)

Emulsion activity was measured by mixing 2 mL of a cell-free supernatant obtained from cells grown in test tubes up to stationary phase (OD_600nm_ = 3.0) with 2 mL of petroleum diesel. The mixture was vortexed at high speed for 2 min. After vortexing, the tubes were left resting for 24 h and the emulsion layer was measured. The emulsion index (*E*_24_) was calculated by dividing the height of the emulsion layer by the total height and multiplying that by 100 [43].

### 4.6. Scanning Electron Microscopy (SEM)

To study the bacterial morphologies of the isolated marine strains, cells were grown in TSB medium to stationary phase (OD_600nm_ = 3.0) and observed by scanning electron microscopy. For this, cells were prepared for SEM, washed with phosphate-buffered saline, and fixed with 2.5% glutaraldehyde at room temperature. Cells were gradually dehydrated in a series of ethanol dilutions and subsequently dried. Samples were then coated with a layer of palladium and examined under 15 kV accelerating voltage in an LEO-420 field emission scanning electron microscope (LEO electron microscopy, Carl Zeiss, S.M.T., Oberkochen, Germany).

### 4.7. Phylogenetic Analysis

Taxonomy was primarily assigned using sequences of 16S rDNA obtained by BLAST for comparison against the NCBI Non Redundant database. The sequences of strains under study and of type strains were aligned using Mega 6.0 program (Philadelphia, PA, USA) from a region of 600 bp approximately, and a phylogenetic tree was built. Phylogenetic distance between sequences was calculated with the neighbor-Joining algorithm and a bootstrap of 1000. Bootstrap values greater than 50% are shown in Figure 1c. *Lactobacillus acidophilus* 16S rDNA sequence was used as outgroup to root the tree.

### 4.8. Genome Sequencing and Sequence Information

Genome sequencing of MS13, MS15a, MS16, and MS19 bacterial strains was performed using Illumina Hiseq sequencing technology (Mr. DNA, Shallowater, TX, USA). Libraries were prepared using KAPA HyperPlus Kits (Kapa Biosystems, Wilmington, MA, USA) by following the manufacturer’s user guide. The concentration of DNA was evaluated using the Qubit^®^ dsDNA H.S. Assay Kit (Life Technologies, Madison, WI, USA) and the average library size was determined using the Agilent 2100 Bioanalyzer (Agilent Technologies, Santa Clara, CA, USA). Libraries were end-sequenced for 500 cycles. *De novo* assembly was carried out using C.L.C. Genomics Workbench version 11.0.1 under quality-filtered reads (Q ≥ 20; no more than two ambiguities; final read length ≥500 bp), with length and similarity cutoffs of 60% and 90%, respectively, and minimum contig length of 5000 bp. Genomes were annotated using the NCBI Prokaryotic Genome Annotation Pipeline version 4.6, released in 2013 and approved on May 16th, 2019 [44]. Gene mining and genomic contexts were visualized using the RAST server [45]. Completeness of genome assemblies was evaluated with BUSCO software version 3.0.2 (Lausanne, Switzerland) and the bacteria subset version odb10 [46].

### 4.9. Identification of Secondary Metabolites, and Detection of Genes Involved in Cypermethrin Catabolism in MS13, MS15a, MS16, and MS19 Bacterial Marine Strains and a Subsequent Comparison with Reference Strains

Biosynthetic gene clusters (BGC) were determined through AntiSMASH bacterial version 5.0 [33] using the assembled genomes of selected isolated strains. The comparison was made with reference strains using NCBI information. Differences (discolored) or similarities (colored) of BGCs were established manually. To identify cypermethrin catabolic genes, specifically *est, pytH,* and *laccase* genes, *in silico* analyses were performed with a search by homology approach using BLAST and the following sequences as probes: AAB61674.1 (*est*), AEV51797.1 (*est*), AEY11370.1 (*pytZ*), RKM76030.1 (*laccase*), ERS87108.1 (*laccase*), ASO18034.1 (*pytH*), and KHF66595.1 (*pytH*). For genomic neighborhoods, five genes upstream and downstream to each *est, pytH,* and *laccase* gene were identified through the nucleotide and protein BLAST database (NCBI, Bethesda, MD, USA).

## 5. Conclusions

This study describes a taxonomic identification and biochemical characterization of cypermethrin-degrading and biosurfactant-producing bacterial strains isolated from a unique and extreme marine ecosystem in the Chilean Northern Patagonia. The availability of MS13, MS15a, MS16, and MS19 genome sequences offers new opportunities for systems metabolic engineering that could be useful for biodegradation of cypermethrin-polluted sediments by biosurfactants-producing microorganisms. 

## Figures and Tables

**Figure 1 marinedrugs-18-00252-f001:**
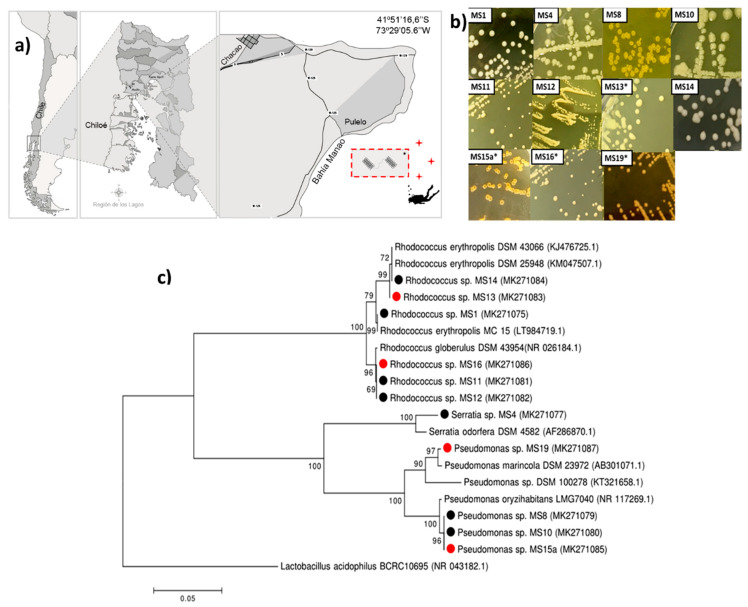
Map of sampling site, geographical position, colony morphologies, phylogeny, and biodiversity of cypermethrin-degrading and biosurfactant-producing bacterial strains isolated from the Chilean Northern Patagonia samples. (**a**) Map of the sampling sites in the Manao Bay: samples were collected from marine sediments at 25 m depth. The asterisk indicates salmon farming center. Crosses mark the three sampling sites. (**b**) Colony morphology, in TSA medium, of cypermethrin-degrading and biosurfactant-producing bacterial strains isolated by enrichment using cypermethrin as the sole carbon and energy source. (**c**) Phylogenetic tree of cypermethrin-degrading and biosurfactant-producing bacterial strains isolated from the Northern Patagonia; black circles indicate cypermethrin-degrading and biosurfactant-producing isolated bacterial strains. Red circles indicate strains selected for genomic analysis.

**Figure 2 marinedrugs-18-00252-f002:**
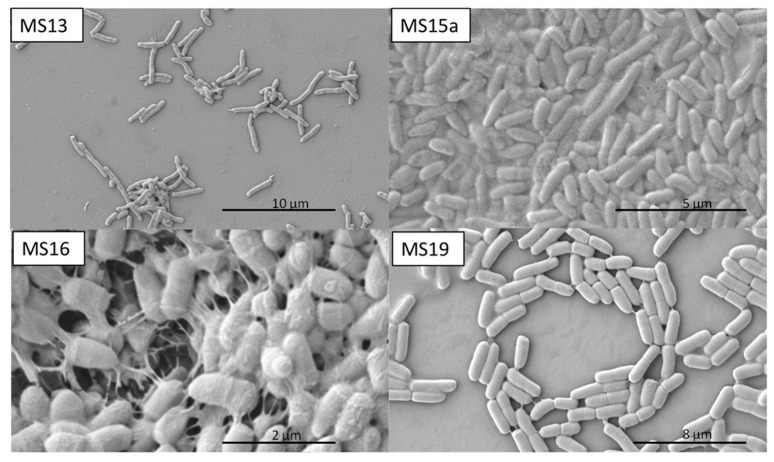
Scanning electron microscopy of cypermethrin-degrading and biosurfactant-producing bacterial strains. Cells tended to occur in pairs. The formation of inter-cellular bonds is highlighted in strain MS16.

**Figure 3 marinedrugs-18-00252-f003:**
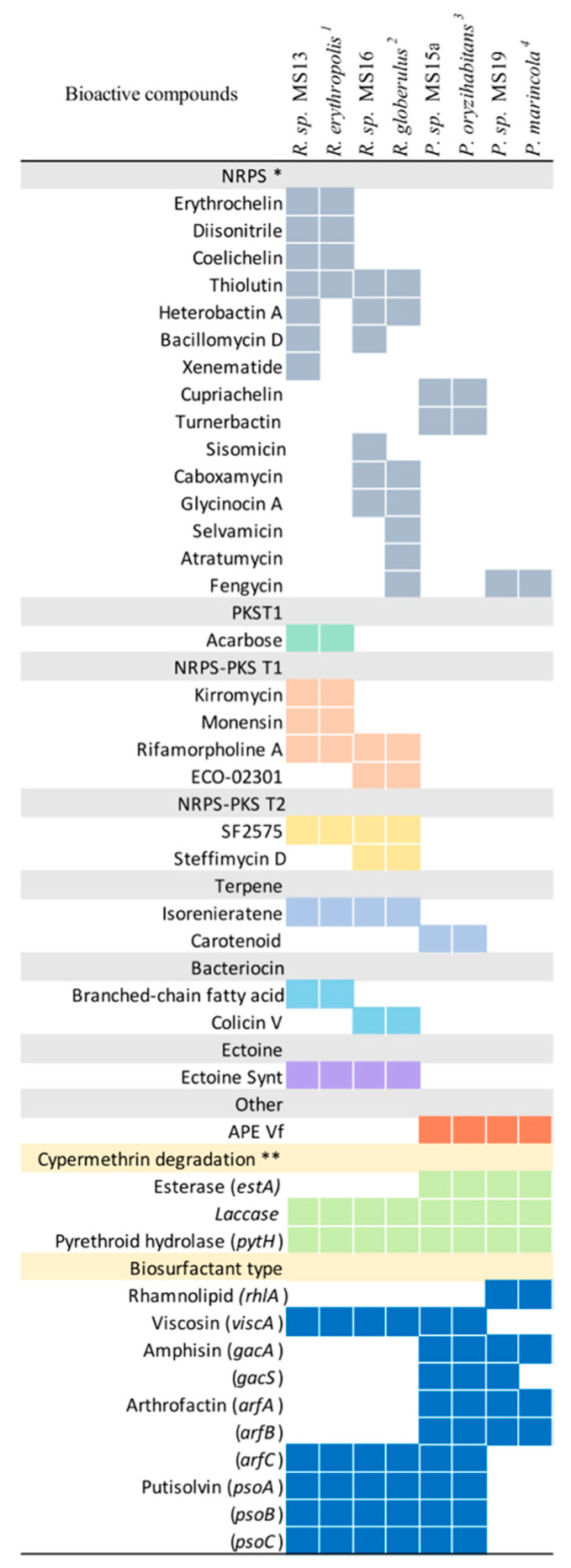
Biosynthetic gene clusters for secondary metabolites and cypermethrin degradation genes in isolated bacteria and reference strains. ^1^ Soil contaminated with fuel. ^2^ Rhizosphere of plants grown in soil contaminated with polychlorinated biphenyls (PCBs). ^3^ Groundwater of pea rhizoplane (*Pisum sativum* L.). ^4^ Deep-sea in the Fiji Sea. * Secondary metabolites identified through antiSMASH5.0 database and ** *in silico* analysis using a search by homology approach.

**Figure 4 marinedrugs-18-00252-f004:**
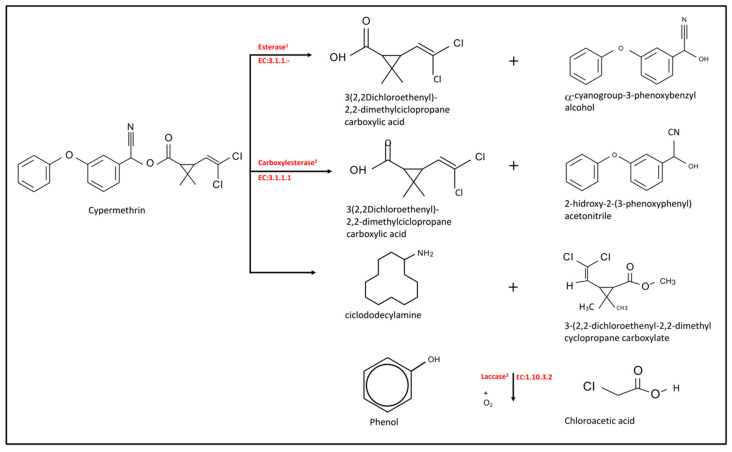
Proposed pathway of cypermethrin biodegradation based on the presence of esterase, laccase, and carboxylesterase-encoding genes in the genomes of MS13, MS16, MS15a, and MS19 isolates. Adapted from ^1^Bhatt et al. [23], ^2^Zhan et al. [24], and ^3^Gangola et al. [17].

**Table 1 marinedrugs-18-00252-t001:** Genome properties and features.

Property	*Rhodococcus* sp. MS13	*Rhodococcus* sp. MS16	*Pseudomonas* sp. MS15a	*Pseudomonas* sp. MS19
Genome size (bp)	6,460,280	6,970,856	5,249,999	4,496,051
N50 (bp)	557,702	745,857	371,249	290,555
G+C content	62%	62%	65%	57%
DNA scaffolds	23	37	27	30
Total genes	5964	6424	4849	4116
RNA genes	62	58	72	67
tRNA genes	54	52	59	56
Pseudogenes	58	158	65	39
Protein-coding genes	5844	6208	4712	4010
Complete BUSCOs	97%	96%	98%	100%

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
