# Peer review of "Biochemical and Genomic Characterization of the Cypermethrin-Degrading and Biosurfactant-Producing Bacterial Strains Isolated from Marine Sediments of the Chilean Northern Patagonia"

_marinedrugs, 2020, doi:10.3390/md18050252_

Round 1
Reviewer 1 Report
The reviewed manuscript present curious results on biochemical and genomic characterization of the cypermethrin-degrading and biosurfactant-producing bacterial strains isolated from marine sediments in Chile. The idea of this study is interesting and has international significance. The manuscript is wery well organized and written, the results are valuable, clearly presented in tables and graphs, and precisely interpreter and discussed.
I would like to draw the Authors' attention to minor editorial shortcomings in the manuscript:
- Line 57. In my wersion something is wrong in temerature record - degree Celsius. Similarly, in lines 252, 266, 271 and 275. Please correct.
- Line 91. After Serratia without dot.
- Fig.1c is hardly legible and clear. Please try to sharpen the image.
- Line 16. The sentence "Cells were grown in TSB medium and observed by scanning electron microscopy" can be ommited in the description of Fig. 2. Such information could be added to the subsection 4.6. Scanning Elecron Microscopy (SEM).
- Line 144. Pseudomonas - should be in italics.
- Line 148. PCBs abbreviation should be explained.
- Lines 155-157. Whether the sentences "Proposed pathway of cypermethrin biodegradation based on the presence of esterases, laccases and carboxylesterases in genome of MS13, MS16, MS15a and MS19 isolates. Adapted from 1Kumar and Sharma, 2020, 2Zhan et al., 2020, Bhatt, 2019 and 3Gangola et al., 2018 should not be worded as follows: Proposed pathway of cepermethrin biodegradation based on the presence of esterases, laccases and carboxylesterases encoding genes in genomes of MS13, MS16, MS15a and MS19 isolates. Adapted from Kumar and Sharma [lack in references], Zhan et al. [37], Bhatt [36] and Gangola et al. [18]?
- Lines 174-175. Choi et al. 2016 should be replaced by Choi et al. [24].
- Line 180. (OD600 nm) should be corrected. Subscript is needed.
- Line 187. Techaoei et al. (2007) should be replaced by Techaoei et al. [26].
- Line 189. Similarly, Borah and Yadov (2016) should be replaced by Borah and Yadov [27].
- Line 203. Undabarrena 2018 [30]. 2018 should be delated.
- Line 213. Ceniceros etal., 2017, and Undabarrena et al., 2018 should be replaced by Ceniceros at al. [30] and Undabarrena et al. [32].
- Line 227. Gongola et al. (2018) should be replaced by Gongola et al. [18].
- Line 238. Kubicki et al. (2019) should be replaced by Kubicki et al. [39].
In conclusion, the manuscript requires minor revision and after correction should be accepted for publishing in Marine Drugs.

Author Response
Dear Editor,
We appreciate the kind suggestions from the reviewers. We have done a meticulous revision of the manuscript in the light of suggestions. We agree that the comments made by the referees have contributed to improving the final version of the manuscript.
We are grateful for your reconsideration of this article for publication, and we hope that this report will fulfill the requirements of Marine Drugs journal. Following your instructions, we resubmit the manuscript, including all the comments of the reviewer and actions taken to correct the manuscript. The answers to the reviewer's questions can be seen below.
Reviewer 1
Line 57. In my version something is wrong in temperature record - degree Celsius. Similarly, in lines 252, 266, 271 and 275. Please correct.
R: Corrected. All the degrees Celsius was replaced.
Line 91. After Serratia without dot.
R: Corrected.
Fig.1c is hardly legible and clear. Please try to sharpen the image.
R: The image was corrected.
Line 16. The sentence "Cells were grown in TSB medium and observed by scanning electron microscopy" can be ommited in the description of Fig. 2. Such information could be added to the subsection 4.6. Scanning Electron Microscopy (SEM).
R: Corrected.
Line 144. Pseudomonas - should be in italics.
R: Corrected.
Line 148. PCBs abbreviation should be explained.
R: Corrected,
Lines 155-157. Whether the sentences "Proposed pathway of cypermethrin biodegradation based on the presence of esterases, laccases and carboxylesterases in genome of MS13, MS16, MS15a and MS19 isolates. Adapted from 1Kumar and Sharma, 2020, 2Zhan et al., 2020, Bhatt, 2019 and 3Gangola et al., 2018 should not be worded as follows: Proposed pathway of cepermethrin biodegradation based on the presence of esterases, laccases and carboxylesterases encoding genes in genomes of MS13, MS16, MS15a and MS19 isolates. Adapted from Kumar and Sharma [lack in references], Zhan et al. [37], Bhatt [36] and Gangola et al. [18]?
R: Corrected. Kumar and Sharma reference was deleted because the correct reference is Bhatt [36].
Lines 174-175. Choi et al. 2016 should be replaced by Choi et al. [24].
R: Replaced.
Line 180. (OD600 nm) should be corrected. Subscript is needed.
R: Corrected.
Line 187. Techaoei et al. (2007) should be replaced by Techaoei et al. [26].
R: Replaced.
Line 189. Similarly, Borah and Yadov (2016) should be replaced by Borah and Yadov [27].
R: Replaced.
Line 203. Undabarrena 2018 [30]. 2018 should be delated.
R: Deleted.
Line 213. Ceniceros etal., 2017, and Undabarrena et al., 2018 should be replaced by Ceniceros at al. [30] and Undabarrena et al. [32].
R: Replaced.
Line 227. Gongola et al. (2018) should be replaced by Gongola et al. [18].
R: Replaced.
Line 238. Kubicki et al. (2019) should be replaced by Kubicki et al. [39].
R: Replaced.
Reviewer 2 Report
This paper describes the isolation and characterisation of bacterial strains that have the ability to degrade cypermethrin and produce biosurfactants. Overall the paper is well written and the approach and methods seem sound.
The only area that there is some room for improvement is the discussion and in-particular putting the results into context.
Questions:
1) There is no discussion on the relative toxicities of the degradation products (for example cyanogroup-3-phenoxybenzyl alcohol, 2,2-dimethylciclopropane carboxylic acid etc are also toxic to aquatic life). What is their relative toxicity when compared to cypermethrin and is there a more desirable degradation metabolic pathway? Could this be optimised?
2) How would these bacteria be used in a bioremediation strategy? Would these bacteria be stimulated in-situ by electron donor addition? Or grown and released into a contaminated area? Or is this a study into natural, passive remediation?
Author Response
Dear Editor,
We appreciate the kind suggestions from the reviewers. We have done a meticulous revision of the manuscript in the light of suggestions. We agree that the comments made by the referees have contributed to improving the final version of the manuscript.
We are grateful for your reconsideration of this article for publication, and we hope that this report will fulfill the requirements of Marine Drugs journal. Following your instructions, we resubmit the manuscript, including all the comments of the reviewer and actions taken to correct the manuscript. The answers to the reviewer's questions can be seen below.
Reviewer 2
1) There is no discussion on the relative toxicities of the degradation products (for example cyanogroup-3-phenoxybenzyl alcohol, 2,2-dimethylciclopropane carboxylic acid etc are also toxic to aquatic life). What is their relative toxicity when compared to cypermethrin and is there a more desirable degradation metabolic pathway? Could this be optimised?
R: Thanks for your comments. We appreciate your kind suggestions, and we include (line 230-234) two references, one of them about toxicity (Chen et al., 2012) and others on relative concentration (Bhatt et al., 2020) of the degradation products like 3-phenoxybenzoic acid. We understand from the reviewer's remark the importance of this point; therefore, we include a sentence indicating the relative toxicity of the cypermethrin degradation products by our bacterial strains should be further investigated (line 230-234). Bhatt et al., 2020
Bhatt P, Huang Y, Zhang W, Sharma A, Chen S. Enhanced Cypermethrin Degradation Kinetics and Metabolic Pathway in Bacillus thuringiensis Strain SG4. Microorganisms. 2020 Feb 7;8(2). pii: E223. doi: 10.3390/microorganisms8020223.
Chen, S.; Luo, J.; Hu, M.; Lai, K.; Geng, P.; Huang, H. Enhancement of cypermethrin degradation by a coculture of Bacillus cereus ZH-3 and Streptomyces aureus HP-S-01. Bioresour. Technol. 2012, 110, 97-104.
(line 246-247), Could this be optimised? There is an approach that made it more efficient the biodegradation of β-cypermethrin and 3-PBA using co-culture of Bacillus licheniformis B-1 and Aspergillus oryzae M-4 [39].
2) How would these bacteria be used in a bioremediation strategy? Would these bacteria be stimulated in-situ by electron donor addition? Or grown and released into a contaminated area? Or is this a study into natural, passive remediation?
R: As was mentioned before, this option needs to be evaluated. Our next step will be evaluated how cypermethrin degrading strains can transform pyrethroid into non-toxic intermediate metabolites. These bacteria first could be used in situ treatment of bioaugmentation like an environmentally sustainable strategy to reduce cypermethrin levels in cypermethrin contaminated sediment. According to Chun et al. [38], the stimulation of PCB dechlorinating and degrading microorganisms with electron-donors/-acceptors addition contributed to the degradation of PCBs in sediment.